# Bloodstream Infections by *Pantoea* Species: Clinical and Microbiological Findings from a Retrospective Study, Italy, 2018–2023

**DOI:** 10.3390/antibiotics12121723

**Published:** 2023-12-13

**Authors:** Roberto Casale, Matteo Boattini, Gabriele Bianco, Sara Comini, Silvia Corcione, Silvia Garazzino, Erika Silvestro, Francesco Giuseppe De Rosa, Rossana Cavallo, Cristina Costa

**Affiliations:** 1Microbiology and Virology Unit, University Hospital City of Health and Science of Turin, 10126 Turin, Italy; 2Department of Public Health and Paediatrics, University of Turin, 10124 Turin, Italy; 3Lisbon Academic Medical Centre, 1649-028 Lisbon, Portugal; 4Operative Unit of Clinical Pathology, Carlo Urbani Hospital, 60035 Jesi, Italy; 5Department of Medical Sciences, Infectious Diseases, University of Turin, 10124 Turin, Italy; 6Infectious Diseases Unit, Department of Pediatric and Public Health Sciences, Regina Margherita Children’s Hospital, 10126 Turin, Italy; 7Unit of Infectious Diseases, Cardinal Massaia Hospital, 14100 Asti, Italy

**Keywords:** *Pantoea* species, bloodstream infection, sepsis, Enterobacterales, Gram-negative, microbiological diagnostics, *Mixta calida*

## Abstract

(1) Background: The widespread use of MALDI-TOF coupled to mass spectrometry has improved diagnostic accuracy by identifying uncommon bacteria. Among Enterobacterales, *Pantoea* species have been seen to be implicated in several human infections, but their clinical and microbiological framework is currently based on a few anecdotal reports. (2) Methods: We conducted this five-year (2018–2023) single-center study aimed at investigating the prevalence and clinical and microbiological findings of *Pantoea* species bloodstream infections. (3) Results: Among the 4996 bloodstream infection Gram-negative isolates collected during the study period, *Pantoea* species accounted for 0.4% (n = 19) of isolates from 19 different patients, 5 of them being pediatric cases. Among *Pantoea* species isolates, *P. agglomerans* was the most frequently detected (45%; n = 9) followed by *P. eucrina* (30%; n = 6) and *P. septica* (15%; n = 3). Malignancy (35.7%) in adults and malignancy (40%) and cerebrovascular disease following meconium aspiration (40%) in pediatric patients as comorbidities and shivering and/or fever following parenteral infusion (36.8%) as a symptom/sign of *Pantoea* species bloodstream infection onset were the most frequently observed clinical features. Among adults, primary bloodstream infection was the most frequent (50%), whereas among pediatric patients, the most commonly identified sources of infection were catheter-related (40%) and the respiratory tract (40%). Overall, *Pantoea* species bloodstream infection isolates displayed high susceptibility to all the antibiotics except for ampicillin (63.2%), fosfomycin (73.7%), and piperacillin/tazobactam (84.2%). Targeted antibiotic treatment was prescribed as monotherapy for adults (71.4%) and combination therapy for pediatric patients (60%). The most prescribed antibiotic regimens were piperacillin/tazobactam (21.4%) in adults and meropenem- (40%) and aminoglycoside-containing (40%) antibiotics in pediatric patients. The overall 28-day all-cause mortality rate was 5.3% (n = 1). (4) Conclusions: The prevalence and 28-day mortality rate of *Pantoea* species bloodstream infections were low. The prescription of targeted therapy including broad-spectrum antibiotics could indicate an underestimation of the specific involvement of the *Pantoea* species in the onset of the disease, warranting further studies defining their pathogenic potential.

## 1. Introduction

Bloodstream infection is a serious clinical condition associated with increased mortality despite the development of new antibiotics and supportive therapies. Gram-negative bacilli are frequently involved in this kind of infection with their effects, which are strongly influenced by the type of pathogen and burden of drug resistance [1], patient conditions and rapidity and appropriateness of diagnosis and treatment [2,3]. Among Enterobacterales, carbapenem-resistant *Escherichia coli* and *Klebsiella pneumoniae* are often the target of surveillance programs and studies, and the literature provides numerous elements on which to base the clinical and diagnostic management of patients with infections sustained by these pathogens [1].

However, the widespread use of MALDI-TOF coupled to mass spectrometry and 16S rRNA gene sequencing have improved diagnostic accuracy in microbiological laboratories, even identifying uncommon bacteria whose clinical relevance and antimicrobial susceptibility are sometimes difficult to determine [4,5], hindering the introduction of early and optimal antibiotic therapy. Furthermore, for many of these bacteria, EUCAST does not provide specific breakpoints, and the results of antimicrobial susceptibility tests must be interpreted using PK-PD (non-species-related) breakpoints.

Among Enterobacterales, the genus *Pantoea* refers to motile, non-capsulated, oxidase-negative, Gram-negative bacilli [6] distributed in environmental habitats such as soil, plants, and feculent material [7] and causing plant infections [8,9]. It includes 20 species, [10] and *P. agglomerans* (formerly *Enterobacter agglomerans*) has been implicated in the majority of the infections in humans as an emerging opportunistic pathogen and causes sporadic infections. In pediatric patients, *Pantoea* species were deemed to be involved in wound infection following penetrating trauma and catheter-related bacteremia [11], postsurgical meningitis [12], bloodstream infection, sepsis or fatal infections in newborns [12,13,14,15,16,17], and nosocomial outbreaks [18,19,20,21]. In adults, wound infections [22,23,24], septic arthritis [25,26,27,28,29,30,31], prosthetic joint infections [32], osteomyelitis [33], bloodstream infections or sepsis [34,35,36,37,38,39,40,41], peritonitis in patients on peritoneal [42,43,44,45,46,47,48] or hemodialysis [49], eye infections [50,51,52,53], endocarditis [54], infection in a near-term female [55], pneumonia post-lung transplant [56], and nosocomial outbreaks [49,57,58,59,60,61] have been reported. *P. dispersa* has been more sporadically reported to cause neonatal sepsis [62], bloodstream infections [63,64,65,66], and rhinosinusitis [67]. To the best of our knowledge, *P. eucrina* and *P. septica* are considered putative pathogens [68], and no infections have been described so far in the literature.

*Pantoea* species’ widespread ecological niche [6,7,8,9] and ability to spread in healthcare settings [18,19,20,21,49,57,58,59,60,61] as well as the progressive increase in the number of immunocompromised and chronically ill patients could represent some favorable conditions for the likelihood of an increasing number of both sporadic and nosocomial infections sustained by these species in the years to come.

Given that our knowledge of infections caused by *Pantoea* species is based on case reports and case series and to inform current antibiotic treatments against these species, we conducted this five-year single-center study aimed at investigating the prevalence and clinical and microbiological findings of *Pantoea* species bloodstream infections.

## 2. Results

Among the 4996 Gram-negative bloodstream infection episodes recorded during the study period, *Pantoea* species accounted for 0.4% (n = 19). Among *Pantoea* species isolates, *P. agglomerans* was the most frequently detected (45%; n = 9) followed by *P. eucrina* (30%; n = 6), *P. septica* (15%; n = 3), and *P. dispersa* (5%; n = 1). These isolates were identified from 19 different patients separated in space and time, 5 of whom were pediatric patients (26.3%).

Adult patients who suffered from *Pantoea* species bloodstream infection had a median age of 68 years [52–78] and pediatric patients less than 1 year [0–2]. They were predominantly male with a median Charlson comorbidity index estimating a 21% probability of adults surviving in the next ten years (Table 1).

Among adults, the most prevalent patient characteristics were malignancy (35.7%) and chronic heart disease (35.7%) followed by diabetes (28.6%), chronic pulmonary disease (28.6%), having been submitted to a surgery within 30 days preceding bloodstream infection onset (28.6%), total parenteral nutrition (14.2%), chronic kidney disease (7.1%), cerebrovascular disease (7.1%), intravenous drug use (7.1%), polytrauma (7.1%), and eating disorder (7.1%). Among pediatric patients, the most prevalent comorbidities were malignancy (40%), cerebrovascular disease following meconium aspiration (40%), total parenteral nutrition (40%), and having been submitted to a surgery within 30 days preceding bloodstream infection onset (20%). Of note, solid neoplasms were prevalent among adults (28.6%), while pediatric patients suffered from hematologic neoplasms (40%).

The median time to bloodstream infection onset from admission was one [0–15] and four days [1–8] for adults and pediatric patients, respectively. Adults and pediatric patients were critically ill in 14.2% and 60% of the cases, respectively. 

Among adults and pediatric patients, *Pantoea* species bloodstream infections were predominantly sustained by *P. agglomerans* (57.1% and 20%, respectively) followed by *P. septica* (21.4% and 60%, respectively) and *P. eucrina* (14.3% and 20%, respectively). They were predominantly healthcare-associated, with 36.8% of patients presenting with shivering and/or fever following a parenteral infusion at bloodstream infection onset. Among adults, primary bloodstream infection was the most frequent (50%) followed by bloodstream infections with an identified source of infection in the central venous line (28.6%), respiratory tract (7.1%), intra-abdominal cavity (7.1%), or soft tissue (7.1%). Among pediatric patients, the most commonly identified sources of infection were catheter-related (40%) and the respiratory tract (40%). In seven cases (36.8%), bloodstream infection was polymicrobial with simultaneous identification of Staphylococcus epidermidis (n = 3), Enterococcus species (n = 2), Lactococcus garvieae (n = 1), and Sphingomonas paucimobilis (n = 1).

Regarding targeted antibiotic treatment, adults were predominantly treated with monotherapy (71.4%), whereas combination therapy (60%) was preferred for pediatric patients. The most frequently prescribed antibiotic regimens were piperacillin/tazobactam (20%) in adults and meropenem- (40%) and aminoglycoside-containing (40%) antibiotics in pediatric patients. Patients with *S. epidermidis or Enterococcus* species polymicrobial bloodstream infection were also treated with vancomycin or daptomycin according to the antibiotic susceptibility testing results. One patient was found to have a Sphingomonas paucimobilis infection and was treated with piperacillin/tazobactam, while Lactococcus garvieae was detected in the same patient with detection of P. eucrina, and no treatment was started.

Regarding outcomes, one adult patient died, contributing to a 28-day all-cause mortality rate of 5.3%. The median length of stay was 10 [7–25] and 15 [10–21] days for adults and pediatric patients, respectively.

Overall, *Pantoea* species bloodstream infection isolates displayed over 90% susceptibility to all the antibiotics except for ampicillin (63.2%), fosfomycin (73.7%), and piperacillin/tazobactam (84.2%) (Table 2). In detail, *P. agglomerans* isolates showed low susceptibility to ampicillin (89%), amoxicillin/clavulanic acid (89%), piperacillin/tazobactam (78%), and fosfomycin (44%). All the *P. septica* isolates displayed resistance to ampicillin and low susceptibility to piperacillin/tazobactam (80%). *P. eucrina* strains displayed low susceptibility to cefotaxime (66.7%), ceftazidime (66.7%), and ertapenem (66.7%), whereas *P. dispersa* was only ampicillin-resistant.

## 3. Discussion

Over the past decade, the widespread use of MALDI-TOF coupled to mass spectrometry has revolutionized clinical laboratory routines, allowing for the implementation of diagnostic protocols aimed at the rapid identification of species and resistance mechanisms [69,70,71,72,73,74,75,76] for early and optimal antibiotic prescription [77]. The increase in speed and diagnostic accuracy has also led to the identification of rare bacterial species whose pathogenic potential has not yet been fully defined. This study offered a contemporary insight into the epidemiology and clinical–microbiological features of *Pantoea* species bloodstream infections, also benefiting from 16S rRNA gene sequencing to avoid misidentification [78]. Its findings reveal *Pantoea* species predominantly caused healthcare-associated bloodstream infections, with a very low prevalence among Gram-negative bacteria. Malignancy in adults and malignancy and cerebrovascular disease following meconium aspiration in pediatric patients as comorbidities and shivering and/or fever following parenteral infusion as a symptom/sign of *Pantoea* species bloodstream infection onset were the most commonly observed clinical features. Microbiologically, despite the multi-susceptibility profile, *Pantoea* species bloodstream infections were predominantly treated with broad-spectrum antibiotics. From a prognostic point of view, the 28-day mortality rate was low, delineating the features of an infection with favorable outcomes.

Comparing the main reported series, *Pantoea* species bloodstream infections are mostly healthcare-acquired [11,37,58,59], being involved in nosocomial outbreaks [18,19,20,21,49,57,61] or otherwise in neonatal, mostly late-onset, sepsis [15,16,17,62]. This finding is consistent with data from our study also showing no community acquisition among pediatric patients, with some cases having been reported, however [14,79].

Immunodeficiency seems to be a dominant clinical feature of patients with *Pantoea* species bloodstream infection, with malignant neoplasms being the main comorbidity in reported series [11,37,58,59,61]. Although completely non-specific, the present study also highlighted the importance of shivering and/or fever following parenteral infusion as a symptom/sign of *Pantoea* species bloodstream infection onset, adding a factor to be evaluated for antibiotic prescription.

*Pantoea* species strains reported to cause bloodstream infections generally displayed a multi-susceptible profile [11,60]. Among pediatric patients, ampicillin might be a reliable treatment option, but resistance to this antimicrobial was largely reported, ranging from 0% [60] to 53.8% [11]. Among adults, extended-spectrum cephalosporins are common treatment options, but resistance rates to these antimicrobials ranged from 0% [60] to 37.7% [11]. Resistance to amoxicillin/clavulanic acid, gentamicin, and fosfomycin was reported sporadically [16,37,79]. Our study highlighted an overall resistance rate to piperacillin/tazobactam of 15.8%, warranting ongoing and vigilant monitoring.

Despite the potential for de-escalation therapy, our study also showed that *Pantoea* species bloodstream infection patients were treated mainly with broad-spectrum antibiotics. On the one hand, this could indicate an underestimation of the specific involvement of *Pantoea* species in the onset of the disease. On the other, it could have been due to both the high Charlson comorbidity index and ampicillin resistance rate. Comparing the main reported series, the mortality of patients with *Pantoea* species bloodstream infection seems to vary quite significantly. In pediatric patients, it ranges from 0% [18] to 12.5% [79], and in adults, from 13% [11] to over 50% [60]. Our study showed a lower all-cause mortality rate among adults (7.1%) and no death among pediatric patients, warranting further multicenter studies to define it properly.

The present study successfully gathered data from a single-center surveillance study, addressing critical gaps in epidemiological, clinical, and microbiological knowledge about *Pantoea* species.

However, some limitations should be acknowledged, including its retrospective nature and the fact that it was conducted in a single center, albeit one with a considerable capacity and encompassing all major medical and surgical specialties. Although our findings probably apply to similar settings, the narrow nature of our results, due to the limited sample size, does not allow generalized conclusions to be drawn. Therefore, the results of antimicrobial susceptibility tests should be interpreted with caution, especially for species represented by a single isolate.

## 4. Materials and Methods

### 4.1. Study Design and Data Collection

In this five-year study (October 2018–October 2023), we included all Gram-negative isolates recovered from positive blood cultures of patients admitted at the “Città della scienza e della salute di Torino”, a 1900-bed tertiary referral hospital in Turin, Northwestern Italy, a region with a high prevalence of multidrug-resistant organisms with complex resistant phenotypes [80]. Duplicate isolates obtained within a 20-day interval from the same patient were considered as part of a single positive blood culture episode and thus excluded from the analysis. The prevalence of *Pantoea* species positive blood culture episodes was investigated. Electronic medical charts of patients who suffered from *Pantoea* species bloodstream infection were retrospectively reviewed and demographic, clinical and microbiological features investigated.

### 4.2. Definitions

A *Pantoea* species bloodstream infection was defined as a bloodstream infection event documented by blood culture positivity for a *Pantoea* strain and concomitant Systemic Inflammatory Response Syndrome signs. Bloodstream infection onset was defined as the collection date of the index BC. Healthcare-associated *Pantoea* species bloodstream infection was defined as infection occurring while the patient was receiving healthcare, infection developed in a hospital or other healthcare facility that first appeared 48 h or more after hospital admission, or infection within 30 days after having received healthcare. The source of *Pantoea* species bloodstream infection was assessed according to the National Healthcare Safety Network, and primary bloodstream infection was defined as not secondary to infection at another body site. Catheter-related bloodstream infection was defined according to criteria of the Infectious Diseases Society of America clinical practice guideline [81].

### 4.3. Microbiological Diagnostics

The BACT/ALERT FA and FN Plus blood culture bottles (bioMérieux, Marcy l’Ètoile, France) were incubated in the BACT/ALERT Virtuo (bioMérieux, Marcy l’Ètoile, France). Flagged positive blood cultures were subjected to Gram staining and subculture on appropriate solid medium (Blood Agar and MacConkey Agar). Microbial species identification was performed on overnight subcultures by using matrix-assisted laser desorption ionization–time of flight mass spectrometry (MALDI-TOF MS, Bruker DALTONIK GmbH, Bremen, Germany). *Pantoea* species identification was validated with a spectral score >2.00 and subsequently confirmed by 16S rRNA gene sequencing. Antimicrobial susceptibility testing was performed through a microdilution method (Panel NMDR on automated Microscan WalkAway 96 Plus System, Beckman Coulter, Nyon, Switzerland), and results were interpreted according to the current EUCAST clinical breakpoints for Enterobacterales (v. 13.1) [82].

### 4.4. Statistical Analysis

Descriptive data are shown as absolute (n) and relative (%) frequencies for categorical data and median and interquartile range (IQR) for continuous variables. Summary statistics for MIC values included the MIC range. Data analysis was performed using Microsoft® Excel® for Microsoft 365 MSO (Version 2311 Build 16.0.17029.20028).

## 5. Conclusions

This study showed both the low prevalence and 28-day mortality rate of *Pantoea* species bloodstream infections. Malignancy in adults and malignancy and anoxic brain injury in pediatric patients as comorbidities and shivering and/or fever following parenteral infusion as a symptom/sign of *Pantoea* species bloodstream infection onset were the most commonly observed clinical features. Despite being predominantly detected in healthcare-associated bloodstream infections, *Pantoea* species showed a profile of antimicrobial multi-susceptibility. The prescription of targeted therapy including broad-spectrum antibiotics could indicate an underestimation of the specific involvement of the *Pantoea* species in the onset of the disease, warranting further multicenter studies for defining their pathogenic potential.

## Figures and Tables

**Table 1 antibiotics-12-01723-t001:** Clinical features of patients with *Pantoea* species bloodstream infection.

	Adults n = 14 % (n)	Paediatric Patients n = 5 % (n)
**Patient characteristics**		
Median age [IQR] (years)	68 [52–78]	0 [0–2]
Male gender	64.3 (9)	60 (3)
Charlson comorbidity index, median [IQR]	5 [2–6]	-
Diabetes	28.6 (4)	-
Malignancy	35.7 (5)	40 (2)
Solid neoplasm	28.6 (4)	-
Hematologic neoplasm	7.1 (1)	40 (2)
Chronic heart disease	35.7 (5)	-
Chronic pulmonary disease	28.6 (4)	-
Chronic kidney disease	7.1 (1)	-
Cerebrovascular disease	7.1 (1)	40 (2) ^1^
Intravenous drug use	7.1 (1)	-
Total parenteral nutrition	14.2 (2)	40 (2)
Polytrauma	7.1 (1)	-
Eating disorder	7.1 (1)	-
Time to bloodstream infection onset from admission (days), median [IQR]	1 [0–15]	4 [1–8]
Critically ill patient	14.2 (2)	60 (3)
Surgery 30 days preceding bloodstream infection onset	28.6 (4)	20 (1)
**Characteristics and source of bloodstream infection**		
*P. agglomerans* etiology	57.1 (8)	20 (1)
*P. septica* etiology	21.4 (3)	60 (3)
*P. eucrina* etiology	14.3 (2)	20 (1)
*P. dispersa* etiology	7.1 (1)	-
Polymicrobial bloodstream infection	42.9 (6)	20 (1)
*Staphylococcus epidermidis*	14.3 (2)	20 (1)
*Enterococcus* species	14.3 (2)	-
*Lactococus garvieae*	7.1 (1)	-
*Sphingomonas paucimobilis*	7.1 (1)	-
Healthcare-associated bloodstream infection	71.4 (10)	100 (5)
Shivering and/or fever following parenteral infusion at bloodstream infection onset	28.6 (4)	60 (3)
Primary bloodstream infection	50 (7)	20 (1)
Catheter-related	28.6 (4)	40 (2)
Respiratory tract	7.1 (1)	40 (2)
Intra-abdominal	7.1 (1)	-
Soft tissue	7.1 (1)	-
**Targeted antibiotic treatment** ^3^		
No treatment	7.1 (1) ^2^	-
Monotherapy	71.4 (10)	40 (2)
Combination therapy	21.4 (3)	60 (3)
Amoxicillin/clavulanate-containing	14.2 (2)	-
Ceftriaxone-containing	14.2 (2)	-
Cefepime-containing	14.2 (2)	-
Ceftobiprole-containing	7.1 (1)	-
Piperacillin/tazobactam-containing	21.4 (3)	40 (2)
Meropenem-containing	14.2 (2)	60 (3)
Aminoglycoside-containing	14.2 (2)	60 (3)
Fluoroquinolone-containing	14.2 (2)	-
**Outcomes**		
Length of stay (days), median [IQR]	10 [7–25]	15 [10–21]
28-day all-cause mortality	7.1 (1)	-

^1^ Anoxic brain injury. ^2^
*P. eucrina*. ^3^ All *Pantoea* species strains were susceptible to the antibiotics used in targeted therapy. Abbreviations: IQR: interquartile range.

**Table 2 antibiotics-12-01723-t002:** Antimicrobial susceptibility of *Pantoea* species strains included in the study.

	*Pantoea* species (n = 19)		*P. agglomerans* (n = 9)		*P. septica* (n = 6)		*P. eucrina* (n = 3)		*P. dispersa* (n = 1)	
	MIC Range (mg/L)	Suscept % (n)	MIC Range (mg/L)	Suscept % (n)	MIC Range (mg/L)	Suscept % (n)	MIC Range (mg/L)	Suscept % (n)	MIC Range (mg/L)	Suscept % (n)
**AMP**	≤4 to >8	63.2 (12)	≤4 to 8	89 (8)	>8	0 (0)	≤4	100 (3)	>8	0 (0)
**AM/CL**	≤8 to >8	94.7 (18)	≤8 to >8	89 (8)	≤8	100 (6)	≤8	100 (3)	≤8	100 (1)
**PTZ**	≤4 to >16	84.2 (16)	≤4 to >16	78 (7)	≤8 to >16	83.3 (5)	≤8	100 (3)	≤8	100 (1)
**CTX**	≤1 to 4	94.7 (18)	≤1 to 2	100 (9)	≤1	100 (6)	≤1 to 4	66.7 (2)	≤1	100 (1)
**CAZ**	≤1 to 8	94.7 (18)	≤1 to 2	100 (9)	≤1	100 (6)	≤1 to 8	66.7 (2)	≤1	100 (1)
**FEP**	≤0.5 to 1	100 (19)	≤0.5 to 1	100 (9)	≤0.5 to 1	100 (6)	≤0.5 to 2	100 (3)	≤0.5	100 (1)
**IPM**	≤1	100 (19)	≤1	100 (9)	≤1	100 (6)	≤1	100 (3)	≤1	100 (1)
**MEM**	≤0.12	100 (19)	≤0.12	100 (9)	≤0.12	100 (6)	≤0.12	100 (3)	≤0.12	100 (1)
**ERT**	≤0.12 to >1	94.7 (18)	≤0.12	100 (9)	≤0.12	100 (6)	≤0.12 to >1	66.7 (2)	≤0.12	100 (1)
**AK**	≤8	100 (19)	≤8	100 (9)	≤8	100 (6)	≤8	100 (3)	≤8	100 (1)
**GM**	≤2	100 (19)	≤2	100 (9)	≤2	100 (6)	≤2	100 (3)	≤2	100 (1)
**CIP**	≤0.06 to 0.25	100 (19)	≤0.06 to 0.25	100 (9)	≤0.06 to 0.25	100 (6)	≤0.06	100 (3)	≤0.06	100 (1)
**LVX**	≤0.5	100 (19)	≤0.5	100 (9)	≤0.5	100 (6)	≤0.5	100 (3)	≤0.5	100 (1)
**CL**	≤2	100 (19)	≤2	100 (9)	≤2	100 (6)	≤2	100 (3)	≤2	100 (1)
**FF**	≤16 to >64	73.7 (14)	≤16 to >64	44 (4)	≤16 to >32	100 (6)	≤16	100 (3)	32	100 (1)
**TMP/SMX**	≤2/38	100 (19)	≤2/38	100 (9)	≤2/38	100 (6)	≤2/38	100 (3)	≤2/38	100 (1)

Abbreviations: Suscept: susceptibility; AMP: ampicillin; AM/CL: amoxicillin/clavulanic acid; PTZ: piperacillin/tazobactam; CTX: cefotaxime; CAZ: ceftazidime; FEP: cefepime; IPM: imipenem; MEM: meropenem; ERT: ertapenem; AK: amikacin; GM: gentamicin; CIP: ciprofloxacin; LVX: levofloxacin; CL: colistin; FF: fosfomycin; TMP/SMX: trimethoprim/sulfamethoxazole.

## Data Availability

The authors confirm that the data supporting the findings of this study are available within the article.

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
