# Peer review of "Bloodstream Infections by Pantoea Species: Clinical and Microbiological Findings from a Retrospective Study, Italy, 2018–2023"

_antibiotics, 2023, doi:10.3390/antibiotics12121723_

Round 1
Reviewer 1 Report
Comments and Suggestions for Authors
The genus Pantoea includes bacteria that can cause opportunistic infections in humans. Pantoea agglomerans is the most commonly reported species responsible for many human infections, including wound infections, bacteremia, pneumonia, and urinary tract infections. Recently, there have been increasing numbers of reports of infections in humans caused by other species of Pantoea, including Pantoea dispersa. The paper presents the results of five years of research on, among others, the prevalence of bacteria of the Pantoea genus and their sensitivity to antimicrobials. The article requires some corrections.
Page 1 lines 35-38 Malignancy in adults and malignancy and cerebrovascular disease following meconium aspiration in infants as well as shivering and/or fever following parenteral infusion as symptom/sign of Pantoea species bloodstream infection onset were the most observed clinical features. - These diseases are not a symptom/sign of Pantoea infection. Pantoea infections may accompany these diseases. Please rearrange the sentence.
Page 1 lines 43-44 Please remove “containing”
Page 2 line 76 Pantoea calida (formerly Mixta calida) - The current name is Mixta calida. The article concerns bacteria of the genus Pantoea, and the species Mixta calida represents a different genus (Mixta genus); therefore, it should be removed from the manuscript. Please correct the manuscript.
​Tambong JT (2019) Taxogenomics and Systematics of the Genus Pantoea. Front. Microbiol. 10:2463. doi: 10.3389/fmicb.2019.02463
Palmer M, Steenkamp ET, Coetzee MPA, Avontuur JR, Chan WY, van Zyl E, Blom J, Venter SN. Mixta gen. nov., a new genus in the Erwiniaceae. Int J Syst Evol Microbiol. 2018 Apr;68(4):1396-1407. doi: 10.1099/ijsem.0.002540. Epub 2018 Feb 27. PMID: 29485394.
Page 4 Table 1 Targeted antibiotic treatment - Please remove ”containing”
Please provide an explanation below the table whether the values presented concern sensitive or resistant strains to the tested antibiotics
Page 4 lines 109-111 Please remove ”containing”
Page 6 lines 141-142 –”Furthermore, we reported the first cases of bloodstream infection caused by P. eucrina and P. septica,….” - Please correct this sentence. The first reports on infection caused by P. eucrina were presented, among others, in the article by Lotte, L., Sindt, A., Ruimy, R. et al. Description of the First Case of Catheter-Related Bloodstream Infection Due To Pantoea eucrina in a Cancer Patient. SN Compr. Clin. Med. 1, 142–145 (2019). https://doi.org/10.1007/s42399-018-0031-6
Page 7 - 4.2 Definitions - This subsection should be placed in the main text rather than in the materials and methods
Page 7 ”……..and subculture on appropriate solid medium.” - What does appropriate solid medium mean? Please provide the name of the substrate used.
Comments on the Quality of English Language"Malignancy in adults and malignancy and cerebrovascular disease following meconium aspiration in infants as well as shivering and/or fever following parenteral infusion as symptom/sign of Pantoea species bloodstream infection onset were the most observed clinical features." - These diseases are not a symptom/sign of Pantoea infection. Pantoea infections may accompany these diseases. Please rearrange the sentence.
Author Response
Turin, 30th November 2023
Editor-in-Chief
Antibiotics
We would like to thank the Editorial Team for his helpful suggestions, which in our view have enhanced the quality and strength of our study. We hope that in this revised version the manuscript is now suitable for publication in Antibiotics.
Please, note that the changes to the original manuscript have been highlighted in the text. The response to the Reviewer’s comments and ensuing modifications in the manuscript are also clearly indicated in the rebuttal.
Comments from Reviewers and point-by-point answers
Reviewer #1:
1) The genus Pantoea includes bacteria that can cause opportunistic infections in humans. Pantoea agglomerans is the most commonly reported species responsible for many human infections, including wound infections, bacteremia, pneumonia, and urinary tract infections. Recently, there have been increasing numbers of reports of infections in humans caused by other species of Pantoea, including Pantoea dispersa. The paper presents the results of five years of research on, among others, the prevalence of bacteria of the Pantoea genus and their sensitivity to antimicrobials. The article requires some corrections.
We thank the Reviewer for these comments.
2) Page 1 lines 35-38 Malignancy in adults and malignancy and cerebrovascular disease following meconium aspiration in infants as well as shivering and/or fever following parenteral infusion as symptom/sign of Pantoea species bloodstream infection onset were the most observed clinical features. - These diseases are not a symptom/sign of Pantoea infection. Pantoea infections may accompany these diseases. Please rearrange the sentence.
We thank the Reviewer for this comment. Symptom/sign refers to shivering and/or fever only. However, the sentence has been revised as follows: "Malignancy in adults (33.3%) and malignancy (40%) and cerebrovascular disease following meconium aspiration (40%) in infants as comorbidities and shivering and/or fever following parenteral infusion (35%) as symptom/sign of Pantoea species bloodstream infection onset were the most observed clinical features.".
3) Page 1 lines 43-44 Please remove “containing”
We thank the Reviewer for this comment. The term was removed.
4) Page 2 line 76 Pantoea calida (formerly Mixta calida) - The current name is Mixta calida. The article concerns bacteria of the genus Pantoea, and the species Mixta calida represents a different genus (Mixta genus); therefore, it should be removed from the manuscript. Please correct the manuscript.
​Tambong JT (2019) Taxogenomics and Systematics of the Genus Pantoea. Front. Microbiol. 10:2463. doi: 10.3389/fmicb.2019.02463
Palmer M, Steenkamp ET, Coetzee MPA, Avontuur JR, Chan WY, van Zyl E, Blom J, Venter SN. Mixta gen. nov., a new genus in the Erwiniaceae. Int J Syst Evol Microbiol. 2018 Apr;68(4):1396-1407. doi: 10.1099/ijsem.0.002540. Epub 2018 Feb 27. PMID: 29485394.
We thank the Reviewer for this accurate appraisal. The case of P. calida was removed and the entire article revised accordingly.
5) Page 4 Table 1 Targeted antibiotic treatment - Please remove ”containing”
We thank the Reviewer for these comment. However, we think that the term should be retained in order to also make explicit the figure of antibiotics used in combination.
6) Please provide an explanation below the table whether the values presented concern sensitive or resistant strains to the tested antibiotics
We thank the reviewer for this comment. Accordingly, the following sentence was added below Table 1: "All Pantoea species strains were susceptible to the antibiotics used in targeted therapy.".
7) Page 4 lines 109-111 Please remove ”containing”
We thank the Reviewer for this comment. However, the term was removed for adults because ceftriaxone and piperacillin/tazobactam were used in monotherapy while it was retained for children as the listed antibiotics were used in combination.
8) Page 6 lines 141-142 –”Furthermore, we reported the first cases of bloodstream infection caused by P. eucrina and P. septica,….” - Please correct this sentence. The first reports on infection caused by P. eucrina were presented, among others, in the article by Lotte, L., Sindt, A., Ruimy, R. et al. Description of the First Case of Catheter-Related Bloodstream Infection Due To Pantoea eucrina in a Cancer Patient. SN Compr. Clin. Med. 1, 142–145 (2019). https://doi.org/10.1007/s42399-018-0031-6
We thank the Reviewer for this accurate comment. Accordingly, the sentence was removed.
9) Page 7 - 4.2 Definitions - This subsection should be placed in the main text rather than in the materials and methods
We thank the reviewer for this comment. We understand the Reviewer's misgivings about the positioning of the subsections on definitions, but the format of the journal foresees the methods at the end of the articles, which in our opinion also makes reading more difficult.
10) Page 7 ”……..and subculture on appropriate solid medium.” - What does appropriate solid medium mean? Please provide the name of the substrate used.
We thank the Reviewer for this comment. The terminology is deliberately generic and widely used in the literature. However, we added the media used for subcultures (Blood Agar and MacConkey Agar).
Reviewer 2 Report
Comments and Suggestions for Authors
The authors provide information about the prevalence of Pantoea bloodstream infection, which has been reported as a rare bacteria associated to bloodstream infections in other reports, highlighting the pathogenic potential that has been underestimated. Because Pantoea has starting to be identify in this type of infections, it results significant to investigate in a bigger cohort and in more then one center.
Only some suggestions:
lines 35-38, sentence is unclear
Define IQR in footnotes
line 173-175, paragraph can be added to the previous one.
lines 187-188, duplicate isolates obtained from the same patient ... this indicates that some of isolates are different isolates from the same patient when the same antibiotic susceptibility testing was observed?
lines 134-142, if P. eucrina was considered as non-pathogenic and no treated, how authors discussed the bloodstream infection was caused by Pantoea?
line 148, grammar
line 169, add comma after children
Results are difficult to see in only one table, maybe if authors split the table may improve to show results and explanations.
Comments on the Quality of English Language
Only some grammar mistakes were detected, however, I am a non-native English speaker, so a proffesional opinion is required.
Author Response
Turin, 30th November 2023
Editor-in-Chief
Antibiotics
We would like to thank the Editorial Team for his helpful suggestions, which in our view have enhanced the quality and strength of our study. We hope that in this revised version the manuscript is now suitable for publication in Antibiotics.
Please, note that the changes to the original manuscript have been highlighted in the text. The response to the Reviewer’s comments and ensuing modifications in the manuscript are also clearly indicated in the rebuttal.
Comments from Reviewers and point-by-point answers
Reviewer #2:
1) The authors provide information about the prevalence of Pantoea bloodstream infection, which has been reported as a rare bacteria associated to bloodstream infections in other reports, highlighting the pathogenic potential that has been underestimated. Because Pantoea has starting to be identify in this type of infections, it results significant to investigate in a bigger cohort and in more then one center.
We thank the Reviewer for this comment.
2) lines 35-38, sentence is unclear
We thank the Reviewer for this comment. The sentence refers to the most observed clinical features observed in patients who suffered from Pantoea species bloodstream infections (please see table 1). The sentence was revised as follows: "Malignancy in adults (33.3%) and malignancy (40%) and cerebrovascular disease following meconium aspiration (40%) in infants as comorbidities and shivering and/or fever following parenteral infusion (35%) as symptom/sign of Pantoea species bloodstream infection onset were the most observed clinical features.".
3) Define IQR in footnotes
We thank the Reviewer for this comment. IQR definition was added in footnotes.
4) line 173-175, paragraph can be added to the previous one.
We thank the Reviewer for this comment. The two paragraphs were merged.
5) lines 187-188, duplicate isolates obtained from the same patient ... this indicates that some of isolates are different isolates from the same patient when the same antibiotic susceptibility testing was observed?
We thank the Reviewer for this comment. This only indicates a selection criterion for trying to define the bloodstream infection episode. However, to try to minimise the risk of interpretation, the sentence has been revised as follows: "Duplicate isolates obtained within a 20-day interval from the same patient were considered as part of a single positive blood culture episode and thus excluded from the analysis.".
6) lines 134-142, if P. eucrina was considered as non-pathogenic and no treated, how authors discussed the bloodstream infection was caused by Pantoea?
We thank the Reviewer for this comment. However, we defined a Pantoea species bloodstream infection as "documented by blood culture positivity for a Pantoea strain and concomitant Systemic Inflammatory Response Syndrome signs" and this was the case for the patient with P. eucrina detection. We consider relevant to report this case both for the methodological criteria of the study and for possible comparisons with future cases where P. eucrina could be found as pathogen.
7) line 148, grammar
We thank the Reviewer for this comment. The sentence was eliminated according to Reviewer 1.
8) line 169, add comma after children
We thank the Reviewer for this comment. Accordingly, comma was added.
9) Results are difficult to see in only one table, maybe if authors split the table may improve to show results and explanations.
We thank the Reviewer for this comment. We provided two tables (1. Clinical features of patients with Pantoea species bloodstream infection; 2. Antimicrobial susceptibility of Pantoea species strains included in the study). We presented a rather rare bloodstream infection as shown by the data on its prevalence. Creating additional tables for such a small number of cases seems disproportionate.
Reviewer 3 Report
Comments and Suggestions for Authors
The authors conducted a 5-year retrospective review of clinical/microbiologic experience with Pantoaea spp BSI. The study investigates a rare GNB associated with infection particularly among immunocompromised hosts and notes some interesting observations including prevalence in children and adults, potential impact of preceding surgery on acquiring infection, and importance of infection prevention given the rate of hospital acquired infection. The methodology is well stated.
Limitations to address:
- Please clarify the definition of CLABSI (line 205-206), was this based on differential time to positivity? clinical suspicion for infection from line with other sources ruled out? The current definition you outlined would include secondary line seeding. Would also recommend using the term CRBSI for clinical line infection as CLABSI is a laboratory/investigational definition.
- Please clarify what is meant by " the most prevalent comorbidity were malignancy (40%) and cerebrovascular disease following meconium aspiration (40%)." on lines 100-101. Is this cerebrovascular disease in context of meconium aspiration? Not sure what that means. Would also clarify this in table 1.
- It may be interesting to highlight what type of malignancy the patients had to see if any particular type (GI vs non-GI) poses a higher risk of Pantoea BSI.
- In the 1 patient that died, was the death due to Pantoea BSI? In a low sample size such as this, it would be prudent to have two independent reviewers evaluate for attributable mortality. This could also be clarified in discussion in lines 170-172.
- The # of Pantoea species identified (n=20) is limited. It is difficult to draw conclusions on susceptibility particularly when <30 isolates. While the table 2 is useful, I'd caution interpretation of susceptibilities especially for species < 5 isolates as even 1 isolate can widely impact susceptibilities. I would emphasize this in limitations.
- In table 1 you highlight that 6 adults and 1 child had polymicrobial bactermia. What were the other polymicrobial pathogens identified? how did this impact treatment?
- It looks like 1 of 15 adults were not treated for Pantaoea BSI. How was pantoea BSI defined as clincially significant? I.e. contaminant vs true infection and how did this process impact decision to treat?
- Among those patients who had surgery preceding Pantoea, what type of surgeries did these individuals have?
- Higher resistance to Ampicillin and fosfomycin was noted in this study similar to prior. Is there any proposed or identified resistance mechanisms that could contribute to this pattern?
- In lines 163-165 authors comment on treatment with oral/monotherapy vs broad spectrum. Did polymicrobial infection contribute to this pattern? I agree that CCI likely played a role, i.e. sicker patients or those with malignancy likely got broad-spectrum, but I can't explain why ampicillin resistance would play a role in narrowing antimicrobials.
- Recommend also noting in limitations that while this is a larger retrospective study, n-20 limits your ability to draw conclusions between antimicrobial susceptibility pattern and clinical efficacy. As noted above, cautious interpretation of AST is also strongly advised as even 1 isolate with a variable susceptibility can dramatically change your %susceptiblity.
- Please highlight/ensure that the IRB approval included both adults and infants as per institutional guidelines?
Citations 4,5, 69, 70 - represent self-citations that are meant to highlight utility of MALDI-TOF in identifying atypical/rare organisms. Would strongly advice referencing the original MALDI-TOF studies for this rather than own research.
Author Response
Turin, 30th November 2023
Editor-in-Chief
Antibiotics
We would like to thank the Editorial Team for his helpful suggestions, which in our view have enhanced the quality and strength of our study. We hope that in this revised version the manuscript is now suitable for publication in Antibiotics.
Please, note that the changes to the original manuscript have been highlighted in the text. The response to the Reviewer’s comments and ensuing modifications in the manuscript are also clearly indicated in the rebuttal.
Comments from Reviewers and point-by-point answers
Reviewer #3:
1) The authors conducted a 5-year retrospective review of clinical/microbiologic experience with Pantoaea spp BSI. The study investigates a rare GNB associated with infection particularly among immunocompromised hosts and notes some interesting observations including prevalence in children and adults, potential impact of preceding surgery on acquiring infection, and importance of infection prevention given the rate of hospital acquired infection. The methodology is well stated.
We thank the Reviewer for these accurate appraisals.
2) Please clarify the definition of CLABSI (line 205-206), was this based on differential time to positivity? clinical suspicion for infection from line with other sources ruled out? The current definition you outlined would include secondary line seeding. Would also recommend using the term CRBSI for clinical line infection as CLABSI is a laboratory/investigational definition.
We thank the Reviewer for this comment. The text was revised and definition of catheter-related bloodstream infection by IDSA considered.
3) Please clarify what is meant by " the most prevalent comorbidity were malignancy (40%) and cerebrovascular disease following meconium aspiration (40%)." on lines 100-101. Is this cerebrovascular disease in context of meconium aspiration? Not sure what that means. Would also clarify this in table 1.
We thank the Reviewer for this comment. Infants who aspirated meconium developed anoxic brain injury and were classified as suffering from cerebrovascular disease. Accordingly, we added a comment as footnote.
4) It may be interesting to highlight what type of malignancy the patients had to see if any particular type (GI vs non-GI) poses a higher risk of Pantoea BSI.
We thank the Reviewer for this comment. Classification of malignancy was added in Table 1.
5) In the 1 patient that died, was the death due to Pantoea BSI? In a low sample size such as this, it would be prudent to have two independent reviewers evaluate for attributable mortality. This could also be clarified in discussion in lines 170-172.
We thank the Reviewer for this comment. We are aware that the limited number of cases is a limitation of this study. At the same time, as showed in Table 1, we investigated the 28-day all-cause mortality rate in patients with Pantoea bsi and not the attributable mortality rate. According to the Reviewer's comment we made it explicit again in the discussion.
6) The # of Pantoea species identified (n=20) is limited. It is difficult to draw conclusions on susceptibility particularly when <30 isolates. While the table 2 is useful, I'd caution interpretation of susceptibilities especially for species < 5 isolates as even 1 isolate can widely impact susceptibilities. I would emphasize this in limitations.
We thank the Reviewer for this comment. Accordingly, the listed topics have been added to the study limitations.
7) In table 1 you highlight that 6 adults and 1 child had polymicrobial bactermia. What were the other polymicrobial pathogens identified? how did this impact treatment?
We thank the reviewer for this comment. The seven cases of polymicrobial bloodstream infection saw identification of Staphylococcus epidermidis (n=3), Enterococcus species (n=2), Lactococcus garvieae (n=1) and Sphingomonas paucimobilis (n=1). Staphylococcus and Enterococcus species were treated with vancomycin or daptomycin according to susceptibility results. Sphingomonas paucimobilis were treated with piperacillin/tazobactam while Lactococcus garvieae was detected in the same patient with P. eucrina and no treatment was started. According to Reviewer's suggestion, isolates detected in polymicrobial bloodstream infections were made explicit in Table 1 and a comment on their targeted antibiotic therapy have been added in the text.
8) It looks like 1 of 15 adults were not treated for Pantaoea BSI. How was pantoea BSI defined as clincially significant? I.e. contaminant vs true infection and how did this process impact decision to treat?
We thank the Reviewer for this comment. We defined a Pantoea species bloodstream infection as "documented by blood culture positivity for a Pantoea strain and concomitant Systemic Inflammatory Response Syndrome signs" and this was the case for the patient with P. eucrina detection. We consider relevant to report this case both for the methodological criteria of the study and for possible comparisons with future cases where P. eucrina could be found as pathogen.
9) Among those patients who had surgery preceding Pantoea, what type of surgeries did these individuals have?
We thank the Reviewer for this comment. The limited number of cases hinders more specific analyses. In the present case, the surgeries performed were of different types (vascular n=1, cardiac n=1, orthopaedic n=1, abdominal n=1, neurosurgery n=1), so we think it is of little use to report them.
10) Higher resistance to Ampicillin and fosfomycin was noted in this study similar to prior. Is there any proposed or identified resistance mechanisms that could contribute to this pattern?
We thank the Reviewer for this comment. The study of resistance mechanisms is beyond the scope of this work, also considering that the species in question are rare and, to date, are not subject to in-depth characterisation.
11) In lines 163-165 authors comment on treatment with oral/monotherapy vs broad spectrum. Did polymicrobial infection contribute to this pattern? I agree that CCI likely played a role, i.e. sicker patients or those with malignancy likely got broad-spectrum, but I can't explain why ampicillin resistance would play a role in narrowing antimicrobials.
We thank the Reviewer for this comment. In our opinion, polymicrobial infections did not have much impact on de-escalation also because they were largely sustained by gram-positives and treated with targeted therapy. With regard to ampicillin, high resistance rate hinders de-escalation to this antibiotic.
12) Recommend also noting in limitations that while this is a larger retrospective study, n-20 limits your ability to draw conclusions between antimicrobial susceptibility pattern and clinical efficacy. As noted above, cautious interpretation of AST is also strongly advised as even 1 isolate with a variable susceptibility can dramatically change your %susceptiblity.
We thank the Reviewer for this comment. Accordingly, the listed topics have been added to the study limitations.
13) Please highlight/ensure that the IRB approval included both adults and infants as per institutional guidelines?
We thank the Reviewer for this comment. IRB approval included both adults and infants.
14) Citations 4,5, 69, 70 - represent self-citations that are meant to highlight utility of MALDI-TOF in identifying atypical/rare organisms. Would strongly advice referencing the original MALDI-TOF studies for this rather than own research.
We thank the Reviewer for this comment. Accordingly, le listed references have been replaced.
Round 2
Reviewer 3 Report
Comments and Suggestions for Authors
Thank you for the modifications and edits made to the manuscript and for adressing all comments appropriately. These are satisfactory and the study should help add to current state of knowledge of Pantoea species BSI.
Author Response
We thank the Reviewer for this comment.